# Attention in Convolutional LSTM for Gesture Recognition

**Liang Zhang**[*]
Xidian University
liangzhang@xidian.edu.cn

**Guangming Zhu**[*]
Xidian University
gmzhu@xidian.edu.cn

**Lin Mei**
Xidian University
l_mei72@hotmail.com

**Peiyi Shen**
Xidian University
pyshen@xidian.edu.cn

**Syed Afaq Ali Shah**
Central Queensland University
afaq.shah@uwa.edu.au

**Mohammed Bennamoun**
University of Western Australia
mohammed.bennamoun@uwa.edu.au

## Abstract

Convolutional long short-term memory (LSTM) networks have been widely used for action/gesture recognition, and different attention mechanisms have also been embedded into the LSTM or the convolutional LSTM (ConvLSTM) networks. Based on the previous gesture recognition architectures which combine the three-dimensional convolution neural network (3DCNN) and ConvLSTM, this paper explores the effects of attention mechanism in ConvLSTM. Several variants of ConvLSTM are evaluated: (a) Removing the convolutional structures of the three gates in ConvLSTM, (b) Applying the attention mechanism on the input of ConvLSTM, (c) Reconstructing the input and (d) output gates respectively with the modified channel-wise attention mechanism. The evaluation results demonstrate that the spatial convolutions in the three gates scarcely contribute to the spatiotemporal feature fusion, and the attention mechanisms embedded into the input and output gates cannot improve the feature fusion. In other words, ConvLSTM mainly contributes to the temporal fusion along with the recurrent steps to learn the long-term spatiotemporal features, when taking as input the spatial or spatiotemporal features. On this basis, a new variant of LSTM is derived, in which the convolutional structures are only embedded into the input-to-state transition of LSTM. The code of the LSTM variants is publicly available[2].

## 1   Introduction

Long short-term memory (LSTM) [1] recurrent neural networks are widely used to process sequential data [2]. Several variants of LSTM have been proposed since its inception in 1995 [3]. By extending the fully connected LSTM (FC-LSTM) to have convolutional structures in both the input-to-state and state-to-state transitions, Shi et al. [4] proposed the convolutional LSTM (ConvLSTM) network to process sequential images for precipitation nowcasting. Thereafter, ConvLSTM has been used for action recognition [5, 6], gesture recognition [7–9] and in other fields [10–12]. When LSTM is used to process video or sequential images, the spatial features of two-dimensional convolutional

---

[*]Equal Contribution
[2]https://github.com/GuangmingZhu/AttentionConvLSTM

neural networks (2DCNN) are generally vectorized before feeding them as input of LSTM [13, 14]. However, the two-dimensional spatial feature maps can be fed into ConvLSTM directly, without the loss of the spatial correlation information. For example, the spatial feature maps of AlexNet/VGG-16 [5, 10] or the spatiotemporal feature maps of three-dimensional CNN (3DCNN) [7, 8] are used as input of ConvLSTM. ConvLSTM was originally proposed to take images as input for precipitation nowcasting, the spatial convolutions are therefore necessary to learn the spatiotemporal features. However, how much do the convolutional structures of ConvLSTM contribute to the feature fusion when ConvLSTM takes as input the spatial convolutional features instead of images? Is it necessary to have different gate values for each element of the feature maps in the spatial domain?

The effect of the convolutional structures in ConvLSTM can be analyzed in three cases. **(a)** ConvLSTM takes original images as input. In this case, the convolutional structures are crucial to learn the spatiotemporal features, as verified in [4]. **(b)** ConvLSTM takes the feature maps of 2DCNN as input. In this case, the effect of the convolutional structures is not always remarkable. Intuitively, the three gates of ConvLSTM can be viewed as the weighting mechanism for the feature map fusion. However, the different gates values for each element of the feature maps in the spatial domain seemingly do not have the function of spatial attention. Therefore, the soft attention mechanism [15] is additionally introduced into the input of ConvLSTM in [5], in order to make ConvLSTM focus on the noticeable spatial features. The improvement (as illustrated in Table 1 of [5]) caused by the attention mechanism on the input can also verify the above claim in some degree. **(c)** ConvLSTM takes the feature maps of 3DCNN as input. Since the 3DCNN networks have learnt the spatiotemporal features, the gates of ConvLSTM are more unlikely to have the function of spatial attention. The last case will be analyzed thoroughly in this paper.

Based on our previous published "3DCNN+ConvLSTM+2DCNN" architecture [8], we construct a preliminary "Res3D+ConvLSTM+MobileNet" architecture and derive four variants of the ConvLSTM component. In the preliminary "Res3D+ConvLSTM+MobileNet" architecture, the blocks 1-4 of Res3D [16] are used first to learn the local short-term spatiotemporal feature maps which have a relatively large spatial size. Then, two ConvLSTM layers are stacked to learn the global long-term spatiotemporal feature maps. Finally, parts of MobileNet [17] are used to learn deeper features based on the learnt two-dimensional spatiotemporal feature maps. The Res3D and MobileNet blocks are fixed, and the ConvLSTM component is modified to derive four variants: **(a)** Removing the convolutional structures of the gates by performing the spatial global average pooling on the input and the hidden states ahead. This means that the convolutional operations in the three gates are reduced to the fully-connected operations. The convolutional structures for the input-to-state transition are reserved to learn the spatiotemporal features. **(b)** Applying the soft attention mechanism to the input (i.e., the feature maps of the Res3D block) of ConvLSTM. **(c)** Reconstructing the input gate using the channel-wise attention mechanism. **(d)** Reconstructing the output gate using the channel-wise attention mechanism.

We do not re-evaluate the cases that ConvLSTM takes as input images or features of 2DCNN in this paper, since the experiments in [4] and [5] can demonstrate the aforementioned claims. We focus on the evaluation of the third case on the large-scale isolated gesture datasets Jester [18] and IsoGD [19], since the "3DCNN+ConvLSTM+2DCNN" architecture was originally proposed for gesture recognition. Experimental results demonstrate that neither the convolutional structures in the three gates of ConvLSTM nor the extra spatial attention mechanisms contribute in the performance improvements, given the fact that the input spatiotemporal features of 3DCNN have paid attention to the noticeable spatial features. The exploring on the attention in ConvLSTM leads to a new variant of LSTM, which is different from the FC-LSTM and ConvLSTM. Specifically, the variant only brings the spatial convolutions to the input-to-state transition, and keeps the gates the same as the gates of FC-LSTM.

## 2   Attention in ConvLSTM

To ensure the completeness of the paper, the preliminary "Res3D+ConvLSTM+MobileNet" architecture is first described. Then, the variants of ConvLSTM are elaborated and analyzed.

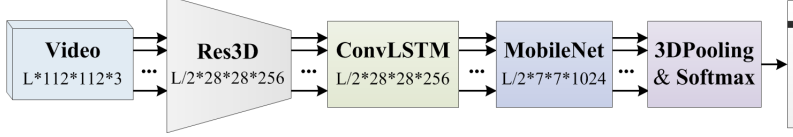

Figure 1: An overview of the "Res3D+ConvLSTM+MobileNet" architecture. The output of each block is in the format of "Length*Width*Height*Channel". MobileNet processes each temporal sample independently.

## 2.1 The preliminary architecture

Two-streams or 3DCNN based networks are widely used for action recognition, such as the famous TSN [20], C3D [21], Res3D [16], and I3D [22] networks. Gesture recognition is different from action recognition. You cannot tell the categories of the dynamic gestures when you only look at an image once. But, you may tell when you just look at an image of actions, under the hints of the backgrounds, objects and postures. Therefore, the aforementioned famous networks cannot produce the state-of-the-art performances on gesture recognition, without including multimodal fusion. Gestures focus on the local information of hands and the global motions of arms. Thus, we use a shallow 3DCNN to learn the local short-term spatiotemporal features first. The 3DCNN block does not need to be deep, since it focuses on the local features. Therefore, the modified blocks 1-4 of Res3D are used. The temporal duration (or spatial size) of the outputted feature maps is only shrunk by a ratio of 2 (or 4), compared with the inputted images. Then, a two-layer ConvLSTM network is stacked to learn the long-term spatiotemporal feature maps. The ConvLSTM network does not shrink the spatial size of the feature maps. Thus, the spatiotemporal feature maps still have a relative large spatial size. The top layers of MobileNet, whose inputs have the same spatial size, are further stacked to learn deeper features. The comparison with the aforementioned famous networks will be given in the experimental part to demonstrate the advantages of the architecture (as displayed in Fig. 1).

## 2.2 The variants of ConvLSTM

Formally, ConvLSTM can be formulated as:

$$i_t = \sigma(W_{xi} * X_t + W_{hi} * H_{t-1} + b_i) \tag{1}$$

$$f_t = \sigma(W_{xf} * X_t + W_{hf} * H_{t-1} + b_f) \tag{2}$$

$$o_t = \sigma(W_{xo} * X_t + W_{ho} * H_{t-1} + b_o) \tag{3}$$

$$G_t = \tanh(W_{xc} * X_t + W_{hc} * H_{t-1} + b_c) \tag{4}$$

$$C_t = f_t \circ C_{t-1} + i_t \circ G_t \tag{5}$$

$$H_t = o_t \circ \tanh(C_t) \tag{6}$$

where $\sigma$ is the sigmoid function, $W_{x\sim}$ and $W_{h\sim}$ are 2-d convolution kernels. The input $X_t$, the cell state $C_t$, the hidden state $H_t$, the candidate memory $G_t$, and the gates $i_t$, $f_t$, $o_t$ are all 3D tensors. The symbol "*" denotes the convolution operator, and "∘" denotes the Hadamard product.

The input $X_t$ has a spatial size of $W \times H$ with $C_{in}$ channels, and ConvLSTM has a convolutional kernel size of $K \times K$ with $C_{out}$ channels. Thus, the parameter size of ConvLSTM can be calculated as[3]:

$$Param_{ConvLSTM} = K \times K \times (C_{in} + C_{out}) \times C_{out} \times 4 \tag{7}$$

The parameter size of ConvLSTM is very large, partly due to the convolutional structures. It can be concluded from Eqs. (1)-(6) that the gates $i_t$, $f_t$, $o_t$ have a spatial size of $W \times H$ with $C_{out}$ channels[4]. It means that the three gates have independent values for each element of the feature maps in the cell state and the candidate memory. In this case, can ConvLSTM focus on the noticeable spatial regions with the help of different gate values in the spatial domain? In order to provide an answer and remove any doubt, four variants of ConvLSTM are constructed as follows (as illustrated in Fig. 2).

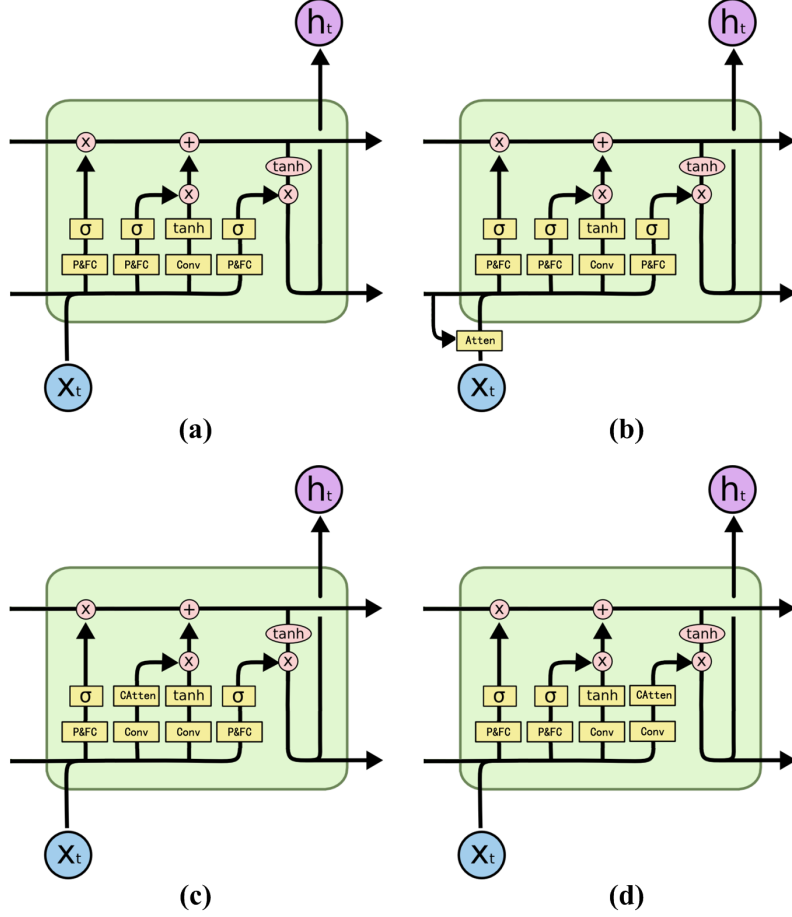

Figure 2: An overview of the four variants of ConvLSTM. The "P&FC" denotes the spatial global average pooling and fully-connected operations, as expressed in Eqs. (8)-(12). The "Conv" denotes the convolutional structure in Eqs. (1)-(4)(13)(17)(21). The "Atten" denotes the standard attention mechanism in Eqs. (17)-(19). The "CAtten" denotes the channel-wise attention in Eqs. (21)-(23).

**(a) Removing the convolutional structures of the gates**

Given the local spatiotemporal features of the 3DCNN block, it can be considered that the 3DCNN block has paid attention to the noticeable spatial regions where there is valuable spatiotemporal information. Therefore, the ConvLSTM block can just focus on the spatiotemporal feature fusion along with the recurrent steps. The gate values are only needed to be calculated for each feature map of the states, not for each element. Therefore, a global average pooling is performed on the input features and the hidden states to reduce the spatial dimension, so that fully-connected operations can be performed instead of convolutions in the gates. The variant of ConvLSTM can be formulated as:

$$\overline{X}_t = GlobalAveragePooling(X_t) \tag{8}$$

$$\overline{H}_{t-1} = GlobalAveragePooling(H_{t-1}) \tag{9}$$

$$i_t = \sigma(W_{xi}\overline{X}_t + W_{hi}\overline{H}_{t-1} + b_i) \tag{10}$$

$$f_t = \sigma(W_{xf}\overline{X}_t + W_{hf}\overline{H}_{t-1} + b_f) \tag{11}$$

$$o_t = \sigma(W_{xo}\overline{X}_t + W_{ho}\overline{H}_{t-1} + b_o) \tag{12}$$

$$G_t = \tanh(W_{xc} * X_t + W_{hc} * H_{t-1} + b_c) \tag{13}$$

$$C_t = f_t \circ C_{t-1} + i_t \circ G_t \tag{14}$$

$$H_t = o_t \circ \tanh(C_t) \tag{15}$$

The gates $i_t$, $f_t$ and $o_t$ are all one-dimensional vectors, so that the elements in each feature map are weighted by the same gate value in Eqs. (14)-(15). The convolutional structures in the three gates are reduced to fully-connected operations. The convolutional structures for the input-to-state transition (as in Eq. (13)) are reserved for the spatiotemporal feature fusion.

In order to reduce the numbers of parameters of the input-to-state transition, the depthwise separable convolutions [23] are used. This reduces the parameter size of the variant of ConvLSTM to

$$Param_{ConvLSTMva} = (K \times K + C_{out} \times 4) \times (C_{in} + C_{out}) \tag{16}$$

Three more variants are constructed based on variant (a), in order to verify whether the spatial attention can improve the performances.

### (b) Applying the attention mechanism to the inputs

By referring to [5], we apply the spatial attention mechanism to the inputs before the operations of Eqs.(8)-(15). Formally, the attention mechanism can be formulated as:

$$Z_t = W_z * \tanh(W_{xa} * X_t + W_{ha} * H_{t-1} + b_a) \tag{17}$$

$$A_t^{ij} = p(att_{ij}|X_t, H_{t-1}) = \frac{\exp(Z_t^{ij})}{\sum_i \sum_j \exp(Z_t^{ij})} \tag{18}$$

$$\tilde{X}_t = A_t \circ X_t \tag{19}$$

where $A_t$ is a 2-d score map, and $W_z$ is the 2-d convolution kernel with a kernel size of $K \times K \times C_{in} \times 1$. The variant (b) can be constructed by replacing $X_t$ in Eqs.(8)-(15) with $\tilde{X}_t$. The parameter size of this variant can be calculated as

$$Param_{ConvLSTMvb} = Param_{ConvLSTMva} + K \times K \times (C_{in} + C_{out} \times 2) + (C_{in} + C_{out}) \times C_{out} \tag{20}$$

### (c) Reconstructing the input gate using the channel-wise attention

Both the gate and the attention mechanisms need to perform convolutions on the input and the hidden states, as expressed in Eqs. (1)-(3) and Eq. (17). Does this mean that the gate mechanism has the function of attention implicitly? The answer is no. The independent gate values in the spatial domain of the feature maps cannot ensure the attention effect as expressed in Eq. (18). Therefore, we reconstruct the input gate according to the attention mechanism. The sigmoid activation function makes the gate values fall in the range 0-1. The division by the sum in Eq. (18) results in attention scores whose sum is 1 in each feature channel. This means that the attention scores in each feature channel may be far less than 1, and far less than most of the normal gate values in other gates, given the large spatial size of the input feature maps. Therefore, the attention mechanism needs to be modified to match the range of the sigmoid function in the gates. Formally, the input gate can be reformulated as:

$$Z_t = W_i * \tanh(W_{xi} * X_t + W_{hi} * H_{t-1} + b_i) \tag{21}$$

$$A_t^{ij}(c) = \frac{\exp(Z_t^{ij}(c))}{\max_{i,j} \exp(Z_t^{ij}(c))} \tag{22}$$

$$i_t = \{A_t^{ij}(c) : (i,j,c) \in \mathbb{R}^{W \times H \times C_{out}}\} \tag{23}$$

where $W_i$ is a 2-d convolution kernel with a kernel size of $W \times H$ and a channel num of $C_{out}$. The "$\max_{i,j} \exp(Z_t^{ij}(c))$" in Eq. (22) corresponds to the maximum element chosen within the channel $c$ of $Z_t$. In other words, the normalization in Eq. (22) is performed channel-wise. The division by the maximum value instead of the sum ensures that the attention scores are distributed in the range of 0-1.

Variant (c) of ConvLSTM can be constructed by replacing the input gate of variant (a) with the new gate expressed by Eqs. (21)-(23). The parameter size of this variant can be calculated as

$$Param_{ConvLSTMvc} = Param_{ConvLSTMva} + K \times K \times (C_{in} + C_{out} \times 2) + C_{out} \times C_{out} \tag{24}$$

**(d) Reconstructing the output gate using the channel-wise attention**

Variant (b) of ConvLSTM applies the attention mechanism on the input feature maps, while variant (c) applies the attention mechanism on the candidate memory. Finally, variant (d) of ConvLSTM is constructed by applying the attention mechanism on the cell state. In other words, the output gate is reconstructed in the same way as the input gate in variant (c). The expressions are similar as in Eqs. (21)-(23), and they are thus omitted for simplicity.

## 3 Experiments

The case in which ConvLSTM takes features from 2DCNN as input has been evaluated in [5], and the improvement (as illustrated in Table 1 of [5]) caused by the attention mechanism on the input features can indicate, in some degree, that the convolutional structures in the gates cannot play the role of spatial attention. Due to page restrictions, this paper only focuses on the evaluation of the case in which ConvLSTM takes features from 3DCNN as input. As aforementioned, the "3DCNN+ConvLSTM+2DCNN" architecture was originally proposed for gesture recognition [8]. Therefore, the proposed variants of ConvLSTM are evaluated on the large-scale isolated gesture datasets Jester [18] and IsoGD [19] in this paper.

### 3.1 Datasets

**Jester**[18] is a large collection of densely-labeled video clips. Each clip contains a pre-defined hand gesture performed by a worker in front of a laptop camera or webcam. The dataset includes 148,094 RGB video files of 27 kinds of gestures. It is the largest isolated gesture dataset in which each category has more than 5,000 instances on average. Therefore, this dataset was used to train our networks from scratch.

**IsoGD**[19] is a large-scale isolated gesture dataset which contains 47,933 RGB+D gesture videos of 249 kinds of gestures performed by 21 subjects. The dataset has been used in the 2016 [24] and 2017 [25] ChaLearn LAP Large-scale Isolated Gesture Recognition Challenges. This paper has the benefit that results are compared with the state-of-the-art networks used in the challenges. Different multi-modal fusion methods were used by the teams in the challenges. In this paper, only the evaluation on each modality is performed (without multi-modal fusion) to verify the advantages of the different deep architectures.

### 3.2 Implementation details

The base architecture has been displayed in Fig. 1. The Res3D and MobileNet components are deployed from their original versions, except for the aforementioned modifications in Section 2.1. These two components are fixed among the variants. The filter numbers of ConvLSTM and the variants are all set to 256.

The networks using the original ConvLSTM or the variants are first trained on the Jester dataset from scratch, and then fine-tuned using the IsoGD dataset to report the final results. For the training on Jester, the learning rate follows a polynomial decay from 0.001 to 0.000001 within a total of 30 epochs. The input is 16 video clips, and each clip contains 16 frames with a spatial size of $112 \times 112$. The uniform sampling with the temporal jitter strategy [26] is utilized to preprocess the inputs. During the fine-tuning with IsoGD, the batch size is set to 8, the temporal length is set to 32, and a total of 15 epochs are performed for each variant. The top-1 accuracy is used as the metric of evaluation. Stochastic gradient descent (SGD) is used for training.

### 3.3 Explorative study

The networks which use the original ConvLSTM or the four variants as the ConvLSTM component in Fig. 1 are evaluated on the Jester and IsoGD datasets respectively. The evaluation results are illustrated in Table 1. The evaluation on Jester has almost the same accuracy except for variant (b). The similar recognition results on Jester may be caused by the network capacity or the distinguishability of the data, because the validation has a comparable accuracy with the training. The lower accuracy of variant (b) may indicate the uselessness of the extra attention mechanism on the inputs, since the learnt spatiotemporal features of 3DCNN have already paid attention to the noticeable spatial regions.

Table 1: Comparison among the original ConvLSTM and the four variants. For simplicity, each row in the column of "Networks" denotes the deep architecture (as displayed in Fig. 1) which takes the original ConvLSTM or its variant as the ConvLSTM component.

| Networks | Validating Accuracy(%) | | Channel Num | Parameter Size | Mult-Adds |
|---|---|---|---|---|---|
| | Jester | IsoGD | | | |
| ConvLSTM | 95.11 | 52.01 | 256 | 4.719M | 3700M |
| Variant (a) | 95.12 | **55.98** | 256 | **0.529**M | **415**M |
| Variant (b) | 94.18 | 43.93 | 256 | 0.667M | 522M |
| Variant (c) | **95.13** | 53.27 | 256 | 0.601M | 472M |
| Variant (d) | 95.10 | 54.11 | 256 | 0.601M | 472M |

The lower accuracy of the variant (b) on IsoGD can also testify this conclusion. The lower accuracy may be due to the additional optimization difficulty caused by the extra multiplication operations in the attention mechanism.

The comparison on IsoGD shows that variant (a) is superior to the original ConvLSTM, regardless of the recognition accuracy or the parameter size and the computational consumption. The reduction of the convolutional structures in the three gates will not reduce the network capacity, but can save memory and computational consumption significantly. The specific attention mechanism embedded in the input and output gates cannot contribute to the feature fusion, but it just brings extra memory and computational consumption. These observations demonstrate that the ConvLSTM component only needs to take full use of its advantages on the long-term temporal fusion, when the input features have learnt the local spatiotemporal information. LSTM/RNN has its superiority on the long sequential data processing. The extension from LSTM to ConvLSTM can only increase the dimensionality of the states and memory, and keep the original gate mechanism unchanged.

This evaluation leads to a new variant of LSTM (i.e., variant (a) of ConvLSTM), in which the convolutional structures are only introduced into the input-to-state transition, and the gates still have the original fully-connected mechanism . The added convolutional structures make the variant of LSTM capable of performing the spatiotemporal feature fusion. The gate mechanism still sticks to its own responsibility and superiority for the long-term temporal fusion.

### 3.4 Comparison with the state-of-the-art

Table 2 shows the comparison results with the state-of-the-art networks on IsoGD. The 2DCNN networks demonstrate their unbeatable superiority on the image-based applications, and also show their ability for action recognition with the help of the specific backgrounds and objects. But, they do not keep their unbeatable performances in the case of gesture recognition, where the fine-grained spatiotemporal features of hands and the global motions of arms do matter. The 3DCNN networks are good at the spatiotemporal feature learning. But, the weakness on long-term temporal fusion restricts their capabilities. The "3DCNN+ConvLSTM+2DCNN" architecture takes full use of the advantages of 3DCNN, ConvLSTM and 2DCNN. The proposed variant (a) of ConvLSTM further enhances ConvLSTM's ability for spatiotemporal feature fusion, without any additional burden. Therefore, the best recognition results can be obtained by taking full use of the intrinsic advantages of the different networks. Although the reference [27] reports the state-of-the-art performance on IsoGD, the high accuracy is achieved by fusing 12 channels (i.e., global/left/right channels for four modalities). The proposed network obtains the best accuracy on each single modality. This exactly demonstrates the superiority of the proposed architecture.

### 3.5 Visualization of the feature map fusion

The reduction of the convolutional structures of the three gates in ConvLSTM brings no side effects to spatiotemporal feature map fusion. Fig. 3 displays an example of visualization of the feature map fusion along with the recurrent steps. It can be seen from the heat maps that the most active regions just reflect the hands' motion trajectories. These are similar to the attention score maps. This also indicates that the learnt spatiotemporal features from 3DCNN have paid attention to the noticeable spatial regions, and no extra attention mechanism is needed when fusing the long-term spatiotemporal

Table 2: Comparison with the state-of-the-art networks on the valid set of IsoGD.

| Deep Architecture | Accuracy(%) | | |
| --- | --- | --- | --- |
| | RGB | Depth | Flow |
| ResNet50 [27] | 33.22 | 27.98 | 46.22 |
| Pyramidal C3D [26] | 36.58 | 38.00 | - |
| C3D [28] | 37.30 | 40.50 | - |
| Res3D [29] | 45.07 | 48.44 | 44.45 |
| 3DCNN+BiConvLSTM+2DCNN[8] | 51.31 | 49.81 | 45.30 |
| Res3D+ConvLSTM+MobileNet | 52.01 | 51.30 | 45.59 |
| Res3D+ConvLSTM Variant(a)+MobileNet | **55.98** | **53.28** | **46.51** |

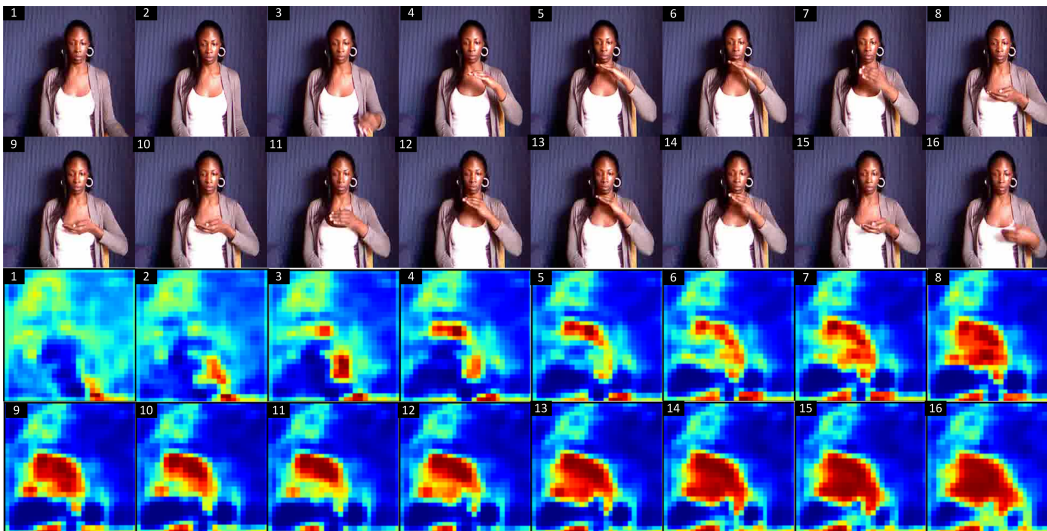

Figure 3: An example of visualization of the feature map fusion in the case of variant (a) of ConvLSTM along with the recurrent steps. The feature map which has the largest activation sum among the 256 channels is visualized.

feature maps using ConvLSTM. The reduction of the convolutional structures of the three gates in ConvLSTM makes the variant more applicable for constructing more complex deep architectures, since this variant has fewer parameters and computational consumption.

## 4 Conclusion

The effects of attention in Convolutional LSTM are explored in this paper. Our evaluation results and previous published results show that the convolutional structures in the gates of ConvLSTM do not play the role of spatial attention, even if the gates have independent weight values for each element of the feature maps in the spatial domain. The reduction of the convolutional structures in the three gates results in a better accuracy, a lower parameter size and a lower computational consumption. This leads to a new variant of LSTM, in which the convolutional structures are only added to the input-to-state transition, and the gates still stick to their own responsibility and superiority for long-term temporal fusion. This makes the proposed variant capable of effectively performing spatiotemporal feature fusion, with fewer parameters and computational consumption.

**Acknowledgments**

This work is partially supported by the National Natural Science Foundation of China under Grant No.61702390, and the Fundamental Research Funds for the Central Universities under Grant JB181001.

## Footnotes

[3]The biases are ignored for simplicity.

[4]It is assumed that the convolutional structures have the same-padding style.

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
