[Reviews · NeurIPS 2018]

Reviewer 1



Summary - The paper explores the effect of attention and convolutional structures in a convLSTM, for the specific case where the ConvLSTM takes spatio-temporal feature maps extracted by a 3D-CNN, and the task is gesture recognition. Experiments are shown on four variants, from which it is discerned that the best architecture is a convLSTM that uses fully connected layers for the gates and convolutional layers for the input-to-state transition, thereby decreasing number of parameters and computation. Strengths - -> Paper is well written, easy to follow, and the figures are good. -> Results obtained using variant a) on gesture recognition are very good, and establish that linear gates are sufficient for spatio-temporal fusion. -> The proposed variant a) significantly reduces number of parameters and computation. Weaknesses - -> The problem statement for attention is slightly shaky. It is evident that feeding spatio-temporal features from a 3D-CNN, would diminish the need for an added attention mechanism inside the convLSTM. The earlier works, like VideoLSTM, exploit attention on convolutional feature maps, which is interesting. This exploration of attention on 3D-CNN feature maps, although rigorous, is not highly interesting or informative, in my opinion. -> The paper shows results for the task of gesture recognition. The proposed architecture could have been adapted for the action recognition task and results shown on the popular UCF and HMDB datasets, to strengthen the arguments put forward in the paper. -> Although relevant papers are discussed in the intro, the paper lacks a comprehensive related work section. Overall assessment - The significance of the paper lies in the newly proposed architecture that reduces number of parameters and computation, but is limited from a research perspective. However, it can add to the knowledge of people working on combing CNN and RNN architectures, as the writing is good and experiments are detailed.

Reviewer 2



This is a form of an ablation study for the case of Conv-LSTMs where the inputs are specifically 3DCNN feature maps. The paper demonstrats that some standard operations of the ConvLSTM module (namely convolutional structure of gates and attention) do not, in fact, contribute meaningfully to learning better models. This is intuitively sensible since the spatiotemporal relationships are already captured in the input features. This is evaluated for gesture recognition. Furthermore, this intuition and experimental verification inspire a new architecture different from ConvLSTM or a fully connected LSTM which retains the performance of Conv LSTM with considerably less parameters. The paper is written clearly. I would advise to keep 8-15 in one block w/o page break for readability. The main reason for the score is that this is practically useful but from a research point view it's a relatively minor contribution as it is a form of relatively straight forward architecture exploration. How is the run time affected by the reduced parameterization? That should be reported in the paper.

Reviewer 3



SUMMARY: This paper proposes a pipeline combining Res3D-3DCNN, convLSTM, MobileNet-CNN hybrid architecture for performing gesture recognition. In particular, it explores the integration of pooling and neural attention mechanisms in ConvLSTM cells. Four convLSTM variants are compared, which place pooling and attention mechanisms at different locations inside the cell. The pooling leads to a somewhat novel LSTM/convLSTM hybrid architecture. Finally, an attention-inspired gating mechanism is proposed, with some differences to the formulation in [5]. The proposed architecture is evaluated on the Jester and ISOGD data sets and achieves state-of-the-art on the RGB-only setting of IsoGD. QUALITY: In the LSTM gates, fully connected weight matrices and matrix multiplication are used. Are there good reasons for doing this? In the original LSTM paper [3], a two vector Hadamard product is proposed (section II.A on the forward pass) instead of matrix multiplication. It will be of interest to compare the matrix-matrix or matrix-vector Hadamard-products in the gates. Following [3] it could also be interesting to explore gating based on output values instead of the cell state. The proposed max-pooling approach should allow this given that same padding was used in the convolutions, so that the spatial resolution inside and outside of the cell should remain the same. Published attention mechanisms have thus far been based on the notion of probability distributions, e.g. in [r6], [r7] or even the cited [5] on which the current attention mechanism is based. However, the proposed "attention" mechanism in equation (22) breaks the probabilistic interpretation by using a max instead of a sum operation. The explanation given in 158-159 is not very clear, why using a max (and not a sum) will distribute the scores between 0-1. Experimentation wise, all proposed modifications have been tested. Additionally a comparison of the attention weights used in variants (c) and possibly a classical soft attention formulation as described in equation (18) would have been interesting in order to enable readers to asses the newly proposed mechanism from eq. (22). Overall the results are state of the art and suffice to support that the authors successfully reduced the number of convLSTM parameters though it is hard to asses the modified attention mechanism proposed by the authors. In particular, looking at the supplementary material results, it looks as if (since the subjects are all stationary and the only movement in the scene is relevant to the hand gesture) that a simple flow detection would suffice. To prove that the attention mechanism is actually working, one would need a baseline comparing against optical flow / movement detection. Also is there some border effects in the attention? There seem to be very strong response in the bottom edge of the scene for almost all samples? Some comparisons to other methods are missing, e.g. the experimental results on the Jester-Data set are not compared to state-of-the-art; they are better than [r4] but worse than [r2]. [r2] reports an accuracy of 96.33% on the Jester V1 data set (using flow and RGB) while [r1] reports an accuracy of 80.96% on the IsoGD-Data set. The paper provides a comparison only to [8], and only to the RGB results, which are significantly worse (51.31% / 58.65% for RGBD). Finally, some statistical analysis would make the results more convincing. CLARITY: The paper is relatively easy to read and follow, though some important details are omitted, e.g. What are the performance metrics being used for evaluation? What variant of gradient descent is used? ORIGINALITY: Attention has been shown to be beneficial in vision problems. The pooling approach is somewhat interesting. Additional related works on gesture recognition are given below in [r1, r2]. Other related works to convLSTM is [r5] which presents a convGRU; an interesting aside is [r3], which found that recurrent dropout may be applied on past state values during gate and candidate value computation. This means that for an RNN to function, past states must not be available in their original values at the gates. This may be why GlobalAveragePooling can be regarded as an interesting choice in the convLSTM case when computing gate values. SIGNIFICANCE: The main contribution of the paper is the experimentation; the paper tests four variants of LSTMs to test the effects of attention in LSTMs. It is found that spatial convolutions in the LSTM gates do not contribute to spatiotemporal feature fusion; furthermore, attention mechanisms embedded into the input and output gates also do not improve the feature fusion. As such, the authors make a recommendation for a new cell architecture which reduces the number of required weights, leading to smaller memory footprints and less training time. Such a cell may also be interesting for those working in action recognition and video understanding. The work adds to the body of literature for anyone working on integrating CNNs and recurrent architures. EXTRA REFERENCES: [r1] Gesture Recognition: Focus on the Hands by Pradyumna Narayana and J. Ross Beveridge in CVPR 2018 [r2] Motion Fused Frames: Data Level Fusion Strategy for Hand Gesture Recognition by Okan Köpüklü and Neslihan Köse and Gerhard Rigoll in CVPR Workshops 2018 [r3] Recurrent Dropout without Memory Loss by Stanislau Semeniuta, Aliaksei Severyn, Erhardt Barth in Proceedings of COLING 2016 [r4] Temporal Relational Reasoning in Videos by Bolei Zhou and Alex Andonian and Antonio Torralba https://arxiv.org/pdf/1711.08496.pdf [r5] Deep Learning for Precipitation Nowcasting: A Benchmark and A New Model by Xingjian Shi et al. in (NIPS 2017) [r6] Listen, Attend and Spell by Chan, William and Jaitly, Navdeep and Le, Quoc V. and Vinyals, Oriol in 2016 IEEE International Conference on Acoustics, Speech and Signal Processing (ICASSP) [r7] Dzmitry Bahdanau, Kyunghyun Cho, and Yoshua Bengio. Neural Machine Translation by Jointly Learning to Align and Translate. In ICLR 2015.